# Encapsulation of catalyst in block copolymer micelles for the polymerization of ethylene in aqueous medium

Camille Boucher-Jacobs [1], Muhammad Rabnawaz[1,3], Joshua S. Katz[2], Ralph Even[2] & Damien Guironnet[1]

The catalytic emulsion polymerization of ethylene has been a long-lasting technical challenge as current techniques still suffer some limitations. Here we report an alternative strategy for the production of semi-crystalline polyethylene latex. Our methodology consists of encapsulating a catalyst precursor within micelles composed of an amphiphilic block copolymer. These micelles act as nanoreactors for the polymerization of ethylene in water. Phosphinosulfonate palladium complexes were used to demonstrate the success of our approach as they were found to be active for hours when encapsulated in micelles. Despite this long stability, the activity of the catalysts in micelles remains significantly lower than in organic solvent, suggesting some catalyst inhibition. The inhibition strength of the different chemicals present in the micelle were determined and compared. The combination of the small volume of the micelles, and the coordination of PEG appear to be the culprits for the low activity observed in micelles.

[1] Department of Chemical and Biomolecular Engineering,, University of Illinois, Urbana-Champaign, Urbana, IL 61801, USA. [2] Formulation Science, Corporate Research and Development, The Dow Chemical Company, Collegeville, PA 19426, USA. [3]Present address: School of Packaging, Michigan State University, 130 Packaging Building, 448 Wilson Road, East Lansing,, MI 48824-1223,, USA. Correspondence and requests for materials should be addressed to D.G. (email: guironne@illinois.edu)

The catalytic polymerization of olefins (primarily ethylene and propylene) accounts for ca. 40% of the annual polymer production worldwide, while the use of emulsion polymerization (primarily with vinyl co-monomers) accounts for ca. 10% of this annual production[1]. Despite the importance of the materials issued from both polymerization processes, these two methods have remained largely incompatible. The extreme water sensitivity of industrial olefin polymerization catalysts (based on early transition metals) makes them unsuitable for emulsion polymerization processes[2,3]. This incompatibility is a great challenge as emulsion polymerization is a very versatile polymerization process resulting in products with unique applications[4–7]. Therefore, there is a strong interest for the development of a strategy enabling the direct synthesis of semi-crystalline polyolefin latexes[8–10].

Olefin polymerization catalyzed by late transition metals offers unique opportunities in regard to emulsion polymerization as these metals are less oxophilic than their early transition metal counterparts and are therefore more stable in the presence of water[11–14]. In 2006, Mecking et al. reported an example of a catalytic polymerization of ethylene in water using water soluble nickel salicylaldimine catalyst percursors[15]. The reaction yielded a stable dispersion of highly crystalline polyethylene (PE) with a particle size of ca. 7 nm on average. Despite the apparent water stability of the catalyst precursor, the productivities of the catalysts in water were lower than in organic solvents. This low activity was attributed to the fast deactivation of the catalyst likely caused by water[16–18]. Since this seminal report, other water-soluble nickel- and palladium-based catalysts have been reported for the polymerization of ethylene in water[15,19–23]. In most cases the activity of the catalysts in water remained significantly lower than the activity in organic solvent. A recent study provides insights on the catalyst activation/deactivation and how to enhance the stability of the catalyst[22]. For instance, performing the polymerization at high pH, adding a weakly coordinating water-soluble labile ligand ($N,N$-dimethylformamide, DMF) and/or replacing the water medium by $D_2O$ resulted in significant improvements in stability. The improved catalyst lifetime provided a most impressive turn over (TO) frequency of 4000 h$^{-1}$ after 24 h of reaction time. Despite this enhancement in productivity and stability, the activity in the aqueous medium remained lower than in organic solvent for the same catalyst (TO ~ 49,800 after 1 h)[24]. Most recently, Mecking et al. reported a series of novel highly electron-deficient nickel (II) salicylaldiminato catalysts exhibiting similar activity in organic and aqueous media[25].

Miniemulsion polymerization has been applied to the polymerization of olefins in aqueous medium[26–37]. In this process, an organic soluble catalyst is dissolved in a highly hydrophobic organic solvent mixture that is then dispersed in nanometer-sized droplets by sonication[38]. These nanodroplets are stabilized by the presence of a large amount of surfactants and the use of a superhydrophobic co-solvent (e.g. hexadecane). Each of these droplets can be described as a small reactor containing an organic solution of the catalyst. Nickel- and palladium-based catalysts have been successfully employed for the polymerization of ethylene in miniemulsion[16,17,22,27,39]. The productivity of these catalysts in miniemulsion remains, however, systematically lower than in organic solvents[28,40].

Overall, the miniemulsion studies and the use of water-soluble catalyst precursors highlight the challenge of performing the catalytic polymerization of ethylene in aqueous media. The difficulty to match the activity achieved in organic solvent with these emulsion polymerization methods motivated us to develop an alternative strategy for the catalytic polymerization of ethylene in aqueous medium.

Inspired by the development of polymeric micelles as nanoreactors, we envisioned applying this technique to perform the catalytic polymerization of ethylene in water[41]. Micelle assemblies offer a unique combination of properties that includes nanometer size control and tunability of the chemistry of the hydrophobic core[41–43]. These nanoscale capsules are made of amphiphilic block copolymers that form micelles in water with a hydrophilic shell and a hydrophobic core potentially loaded with chemical reagents. This methodology has been implemented for a variety of applications including for example: enzymatic catalysis[44–48], radical polymerization (RAFT)[49–51], and artificial organelles[52–54]. More closely related to our approach, Lipshutz et al. reported several examples of successful Pd and Ru catalyzed reactions at room temperature in water using this nanoreactor approach for the synthesis of small molecules (such as alkene metathesis, nucleophilic aromatic substitution, Heck and Suzuki−Miyaura coupling)[55–58].

We herein describe how we implement this encapsulation strategy for the catalytic polymerization of ethylene in micelles. An in-depth analysis of the polymerization results leads us to rationalize the systematically low polymerization rate observed in micelles.

## Results

**Encapsulation strategy**. The concept is schematically shown in Fig. 1.

**Catalyst system**. We focus our investigation on a family of phosphinosulfonate palladium catalysts because these catalysts were shown to exhibit high activity in organic solvent and to be mostly inactive in miniemulsion or as a water-soluble catalyst[39,59,60]. Two ligands ($L_1$ and $L_2$) and three labile ligands (X) were used, resulting in the formation of four distinct catalyst precursors (Fig. 2).

A series of polymerizations were performed to first confirm the high activity of the catalysts in toluene and second to probe their water sensitivity. While the polymerization of ethylene in toluene by $L_1Pd-NR_3$ resulted in activity of ca. TO frequency = 204,000 h$^{-1}$ (Table 1, entry 1), three separate miniemulsion polymerizations with this catalyst precursor and a polymerization in water initiated by a water-soluble catalyst precursor resulted in TO frequency of less than 50 h$^{-1}$ (Table 1, entries 2 and 3). Interestingly, however, when the polymerization was performed in a biphasic toluene/water mixture (10% water), the catalyst activity only decreased by a factor of two suggesting that the catalyst is only moderately sensitive to water (121,000 h$^{-1}$, Table 1, entry 4). Polymerizations performed in this biphasic solvent mixture for various times showed that the yield increased linearly with time suggesting that water inhibits the polymerization but does not cause any apparent deactivation under these conditions (Table 1, entries 4−6)[40]. Similar polymerization conditions with $L_2Pd-DMSO$ showed that this catalyst is slightly more inhibited by water, as its activity in wet toluene was only a third of what it was in dry toluene (Table 1, entry 8 vs. entry 7). Nonetheless, these high activities in organic solvent and moderate water sensitivity make these catalysts of particular interest for the production of semi-crystalline PE latex.

**Block copolymer**. A series of amphiphilic diblock copolymers was synthesized and used to make spherical micelles in water. The diversity of chemical functionality of the different blocks was limited by the fact that the block copolymer (BCP) and the catalyst have to be soluble in the same organic solvent (vide supra)[41]. We, therefore, used exclusively polyethylene glycols (PEG) for the hydrophilic part of the BCP. Three monomers were used for the synthesis of the hydrophobic block. The low polarity of these

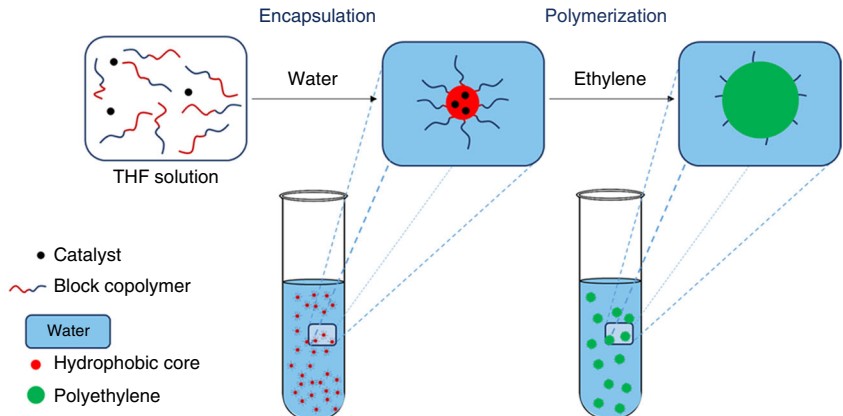

**Fig. 1** Catalyst encapsulation approach for ethylene polymerization in water. Our technical approach consists of first, encapsulating an olefin polymerization catalyst in micelles formed by an amphiphilic block copolymer and second, performing the polymerization of ethylene within these micelles to yield a stable PE latex

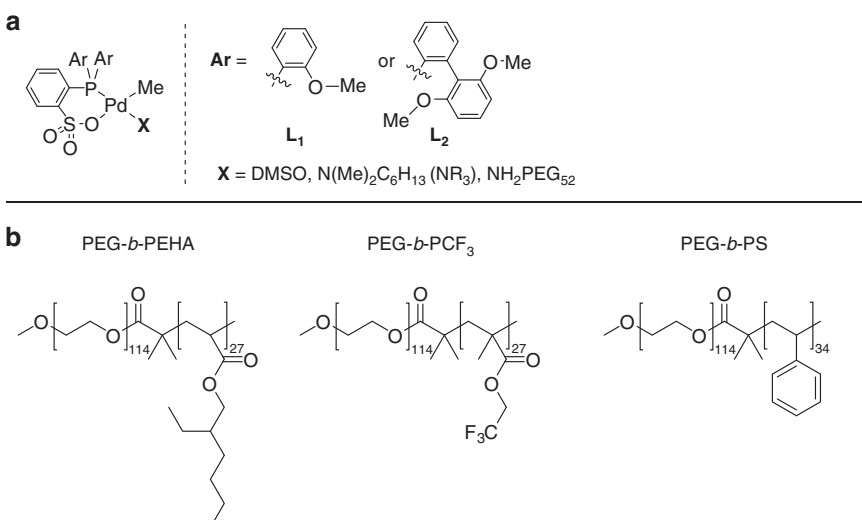

**Fig. 2** Catalysts and block copolymers structure. **a** Catalysts: $L_1$Pd-X and $L_2$Pd-DMSO catalysts. **b** Block copolymers: PEG-*b*-PEHA, PEG-*b*-PCF$_3$, PEG-*b*-PS

monomers is thought to exclude water from the core of the micelles and thus protect the catalyst from water inhibition. Another design criteria for the block copolymer was to exhibit the lowest water solubility possible. This low solubility would decrease the exchange between BCP dissolved in water and BCP forming micelles and therefore limit the potential exposure of the catalyst toward water[43,61].

The diblock copolymers were synthesized by first converting a monomethyl ether PEG-OH (5 kg mol$^{-1}$) into an ATRP macro-initiator, which was then used to perform the living radical polymerization of the different vinyl monomers[62]. In all cases, the molecular weight of the hydrophobic block was targeted to be roughly the same as the molecular weight of the hydrophilic block ($\sim$ 5000 g mol$^{-1}$) in order to favor the formation of spherical micelles[61,63].

PEG-*b*-PEHA (PEHA: poly(ethyl)hexylacrylate), PEG-*b*-PCF$_3$ (PCF$_3$: poly(2,2,2-trifluoroethyl methacrylate)) and PEG-*b*-PS (PS: polystyrene) were successfully synthesized as illustrated by the narrow and monomodal molecular weight distribution measured by GPC (Fig. 2, Supplementary Table 1).

**Micelle formation.** In order to encapsulate the hydrophobic catalyst within the micelle, the formation of micelles has to be done via an indirect method in which the catalyst and block copolymer are both dissolved in a water-miscible organic solvent (e.g. tetrahydrofuran or dimethyl formamide) before the slow addition of water[64–66]. Upon this addition, the block copolymer aggregates to form stable micelles. Dynamic light scattering (DLS) measurement was used to determine the size and distribution of the micelles formed. The ability of the block copolymer to form homogeneous micelles reproducibly was tested prior to loading the micelle with the catalyst. The transparent appearance of the micellar solution provides the first evidence for the formation of homogeneous micelles as a good micellization should result in a clear solution (no visible light diffraction). The synthesized block copolymers (PEG-*b*-PEHA, PEG-*b*-PS, and PEG-*b*-PCF$_3$) resulted in the formation of transparent micelle solutions. PEG-*b*-PS and PEG-*b*-PCF$_3$ resulted in uniform and narrow micelle size distributions. The DLS measurement of the PEG-*b*-PEHA micelle solution showed a single population with a broad size distribution (Supplementary Fig. 9) and therefore was not further investigated. Taking in account the glass transition temperatures of

**Table 1 Catalytic polymerizations of ethylene by palladium catalyst in different reaction media**

| Entry | Solvent | Catalyst [μmol] | Time [min] | Yield PE [g] | TO.h$^{-1a}$ | $M_n$(by GPC)[g mol$^{-1}$] |
|---|---|---|---|---|---|---|
| 1 | Toluene | L$_1$Pd-NR$_3$ (0.9) | 30 | 2.6 | 204,000 | 13,670 |
| 2[b] | Miniemulsion | L$_1$Pd-NR$_3$ (18) | 60 | <0.05 | <50 | n.d. |
| 3[c] | Water | L$_1$Pd-NH$_2$PEG (15) | 60 | 0.02 | 55 | n.d. |
| 4 | 9/1 Toluene/H$_2$O | L$_1$Pd-NR$_3$ (0.9) | 30 | 1.5 | 121,000 | 9750 |
| 5 | 9/1 Toluene/H$_2$O | L$_1$Pd-NR$_3$ (0.9) | 60 | 2.8 | 105,655 | n.d. |
| 6 | 9/1 Toluene/H$_2$O | L$_1$Pd-NR$_3$ (0.9) | 90 | 3.8 | 98,513 | n.d. |
| 7 | Toluene | L$_2$Pd-DMSO (0.4) | 15 | 2.0 | 774,000 | 96,330 |
| 8 | 9/1 Toluene/H$_2$O | L$_2$Pd-DMSO (0.4) | 15 | 0.65 | 226,000 | 68,050 |

*Reaction conditions*: 85 °C, 40 bar of constant ethylene pressure, total volume of solvent: 100 mL *n.d.* not determined
[a]Mol of ethylene consumed per mol of Pd per hour
[b]Average size of micelles or miniemulsion before (78.5 nm) and after (82.4 nm) ethylene polymerization determined by DLS
[c]With additional 800 mg Tergitol® 15-S-20

polystyrene and poly(2,2,2-trifluoroethyl methacrylate), PEG-*b*-PCF$_3$ was chosen as the main amphiphilic block copolymer for this study (lower Tg). PEG-*b*-PCF$_3$ provided micelles with an average particle size of 26 nm. The thermal stability of the micelle solution was probed by maintaining the micelles at 85 °C under vigorous stirring (ethylene polymerization temperature). After 1 h at that temperature and subsequent cooling to room temperature, the micelle size and homogeneity remained mostly unchanged (Supplementary Table 2, entry 2). Dilution of the micelle solution did not result in any significant change in size of the particle which led us to conclude that PEG-*b*-PCF$_3$ BCP forms a thermodynamically stable micelle in water solution and not kinetically trapped metastable nanoparticles.

The loading of the catalyst into the micelles was achieved by dissolving it in the organic solution containing the block copolymer prior to adding the water. In our first attempt we tried to encapsulate L$_1$Pd-DMSO. Upon addition of the water, the solution of L$_1$Pd-DMSO and PEG-*b*-PCF$_3$ became turbid, suggesting that the catalyst was not successfully encapsulated in the micelles or that the micelles did not form homogeneously. We replaced DMSO by a more hydrophobic labile ligand, *N,N*-dimethylhexylamine (L$_1$Pd-NR$_3$). This choice was motivated by the fact that tertiary amines are known to be relatively labile and thus L$_1$Pd-NR$_3$ was expected to have similar activity to L$_1$Pd-DMSO. This change in the ligand modified the solubility of the pre-catalysts that resulted in the formation of a clear micellar solution. The absence of precipitate and DLS measurement confirmed the successful encapsulation with the detection of a single population of narrowly dispersed micelles with an average size of 25 nm. Despite the absence of any evidence for agglomerate formation by DLS measurement, the micelle solutions were systematically filtered using a syringe filter with a pore size of 450 nm and only negligible quantities of residue were collected on the filter (mass loss <1%).

**Ethylene polymerization in micelles**. Micelles made of PEG-*b*-PCF$_3$ loaded with L$_1$Pd-NR$_3$ were used to catalyze the polymerization of ethylene in emulsion. In our first experiment, the reactor was loaded with a 100 mL aqueous micelle solution containing 430 mg of PEG-*b*-PCF$_3$ and 16 μmol of L$_1$Pd-NR$_3$, pressurized under 40 atm of ethylene and the temperature set to 85 °C. After 1 h a milkish solution was collected and no coagulate was observed in the reactor. The formation of polyethylene particles was first confirmed by DLS analysis. The latex was composed of a single population of particles with an average diameter of 84 nm (Table 2, entry 2). The monomodal particle size distribution suggested that the polymerization indeed occurred in the micelles. The activity of the catalyst, determined

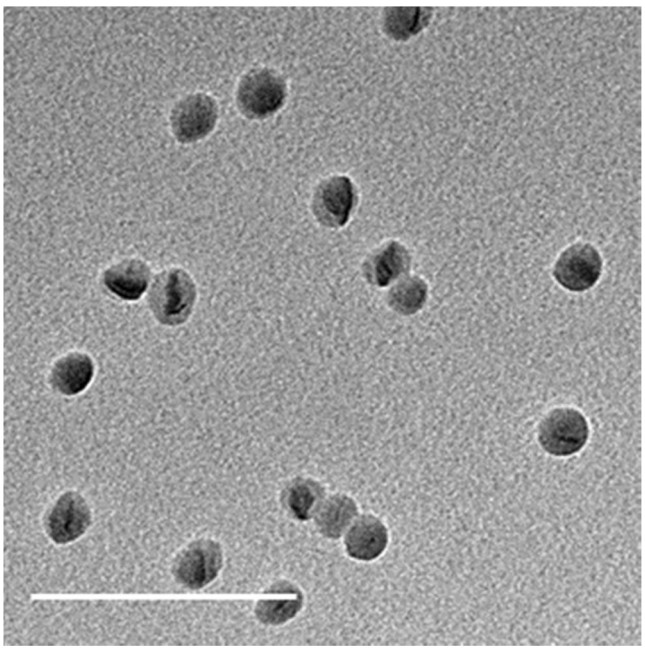

**Fig. 3** TEM image. TEM image of polyethylene particles synthesized by encapsulating L$_1$Pd-NR$_3$ in micelles of PEG-PCF$_3$ and pressurizing with ethylene for 1 h at 40 bar and 85 °C (Table 2, entry 2). Scale: 500 nm. Additional images available in SI (Supplementary Fig. 13, images a and b)

by gravimetric analysis of the latex solution, was 820 TO h$^{-1}$ (Table 2, entry 2). This activity represents a 15-fold improvement in comparison to the miniemulsion polymerization under similar catalyst loading, temperature, and time (Table 1, entry 2).

TEM images of latexes samples were obtained and confirmed the size of the particles and their homogeneity measured by DLS (Fig. 3 and Supplementary Fig. 13 Supporting Information). The particles appear to be very similar to the previously reported particles obtained by miniemulsion polymerization, with lentil-like shapes and no coalescence at their boundaries[7].

Variation in catalyst and block copolymer loadings (Table 2) showed that catalyst concentration in the micelles correlates directly with the size of the polyethylene particle formed. Higher catalyst concentration achieved either by an increase in catalyst loading or by lower block copolymer loading resulted systematically in the formation of larger particles. This is important as it confirms that the polymerization starts in the micelles and then continues in the particles. Catalyst and block copolymer loadings

| Entry | Catalyst loading [μmol] | BCP PEG-*b*-PCF₃ [mg] | Yield PE [g] | TO.h⁻¹ ᵃ | $M_n$ (by GPC) [g mol⁻¹] | Đ | PSᵇ (before) [nm] | PSᵇ (after) [nm] |
|---|---|---|---|---|---|---|---|---|
| 1 | 29 | 430 | 0.58 | 720 | 3590 | 1.3 | 24 | 117 |
| 2 | 18 | 430 | 0.40 | 820 | 1330 | 1.7 | 26 | 84 |
| 3 | 9 | 430 | 0.21 | 830 | 2840 | 1.5 | 26 | 57 |
| 4ᶜ | 18 | 250 | 0.30 | 610 | -ᵈ | -ᵈ | 26 | 154 |
| 5ᶜ | 18 | 750 | 0.34 | 700 | 1490 | 2.6 | 25 | 68 |

**Table 2 Polymerization results for various catalyst (L₁Pd-NR₃)/BCP (PEG-*b*-PCF₃) ratios**

*Reaction conditions*: catalyst: L₁Pd-NR₃, 85 °C, 40 bar of constant ethylene pressure, 1 h, total volume of solvent: 100 mL
ᵃMol of ethylene consumed per mol of Pd per hour
ᵇAverage particles size by volume before and after ethylene polymerization determined by DLS (See Supplementary Table 3)
ᶜRatio THF/PEG-b-PCF₃ maintained constant by varying the THF volume
ᵈMolecular weight below detection limit of GPC

had some moderate effect on the catalyst activity. The constant TOF per Pd at various block copolymer and catalyst loading suggests that the polymerization is not mass transfer limited. Micelles made from 430 mg of PEG-*b*-PCF₃ and 18 μmol of L₁Pd-NR₃ were identified to provide one of the highest activities. The activity in micelles, however, remained two orders of magnitude lower than the activity observed for this catalyst in organic solvent; this lower activity was initially attributed to fast catalyst decomposition.

The stability of the catalyst was probed by performing a series of polymerizations for various times. To our surprise, the catalyst showed negligible decomposition within the first 90 min of the polymerization as illustrated by the linear increase in TO versus time (Fig. 4a). The 3-h polymerization showed some decay in activity but this polymerization also resulted in coagulate formation likely due to the inability of the block copolymer to sufficiently stabilize the larger polyethylene particles formed. (The TO reported does not include the coagulum.) The stability of the catalyst in micelles for several hours while being 100 times less active than in organic solvent suggests that inhibition occurs inside the micelles. The difference in activity was also accompanied by the formation of polyethylene with lower molecular weight. This implies that the ratio of the rate of chain transfer to chain propagation is somehow larger in micelles than in organic solvent. This observation led us to postulate that a chemical present in the micelle could be competing with the monomer for coordination to the metal resulting in a lower rate of propagation without affecting the rate of β-hydride elimination (presumably zeroth order in ethylene)[60]. As an additional element supporting inhibition as the limiting factor of the catalyst activity in micelles, polymerizations at various ethylene pressures (30, 40, and 50 bar) and thus higher concentration in micelles showed that increasing the pressure results in a linear increase of the activity of the catalyst (Fig. 4b). In comparison we found that ethylene saturation in toluene was reached at 10 bar with a constant TO frequency at 10, 30, 40, and 50 bar[60,67,68] (Supplementary Fig. 14).

**Catalyst inhibition.** The catalyst in micelle is exposed to a significantly different environment than in toluene. Molecules such as water, BCP (ester and ether functionalities), and THF will compete with ethylene for coordination to the metal center resulting in lower rates of polymerization. With the assumption of a fast and reversible pre-equilibrium, Eq. 2 (Fig. 5) describes the polymerization rate. Further derivation results in Eq. 3 that demonstrates that the inverse of the catalyst activity (TOF) is linearly dependent on the concentration of inhibitor [L] present in solution (see Supplementary Information for full derivation). The intercept of this regression corresponds to the inverse of the

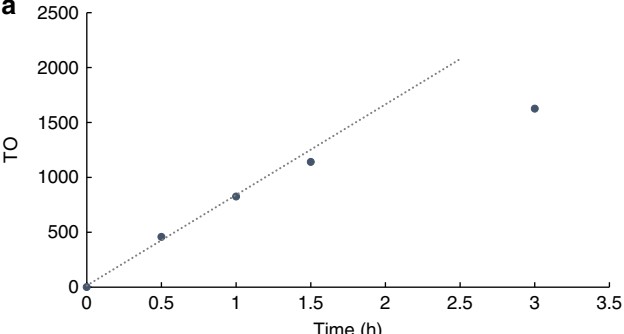

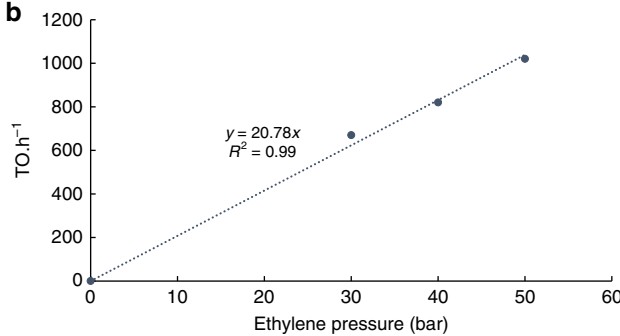

**Fig. 4** Catalyst activity in micelles **a**) Plot of TO over time (micelles formed with: 16 μmol L₁Pd-NR₃, 430 mg PEG-*b*-PCF₃, at 85 °C, 40 bar ethylene) Linear regression includes 0 h, 0.5 h and 1 h runs. **b**) Plot of TO frequency at various ethylene pressures (micelles formed with: 16 μmol L₁Pd-NR₃, 430 mg PEG-b-PCF₃, at 85 °C, 1 h) showcasing that the catalyst's rate remains dependent on ethylene concentration in the micelles

rate constant of propagation of the catalyst and the slope of this regression can be used to determine the inhibition strength ($K_{eq}$) of the labile ligand L (Eq. 3)[40,69]. The $K_{eq}$ of each of the chemicals listed above as well as the rate constant of propagation of the catalyst were determined by performing a series of polymerizations in toluene in presence of various amount of inhibitor (L) at constant ethylene concentration and temperature (see SI, Supplementary Figs. 3—8). The narrow distribution of the rate constant of polymerization of the catalyst $k_p = 197,810 \pm 3.3\%$ confirms the validity of the fast pre-equilibrium assumption. From this data, presented in Table 3, we concluded that water and THF are unlikely to be the cause of the low activity in micelle. The strong amine coordination was also determined to be more inhibiting in the micelle than in the organic solution. Indeed the catalyst is assumed to stay exclusively within the hydrophobic

$$K_{eq} = \frac{[2][L]}{[1][C_2H_4]}$$

$$TOF = \frac{R_p}{[Pd]_0} = \frac{R_p}{[1]+[2]}$$

$$R_p = k_p[2] \qquad (1)$$

$$R_p = K_{eq}k_p\frac{[1][C_2H_4]}{[L]} \qquad (2)$$

$$\frac{1}{TOF} = \frac{1}{K_{eq}k_p[C_2H_4]}[L] + \frac{1}{k_p} \qquad (3)$$

**Fig. 5** Derivation of rate of polymerization assuming a fast and reversible pre-equilibrium. Equation (3) highlights the linear dependency of the inverse TOF as a function of the concentration of L a coordinating agent competing with ethylene. The slope of this line provides information on the equilibrium constant and rate constant of polymerization of the catalyst

**Table 3 Determination of inhibition constant**

| Entry[a] | Chemical | $K_{eq}$ | $k_p$ |
|---|---|---|---|
| 1 | Water | 7.9 E-2 | 204 650 |
| 2 | THF | 7.0 E-1 | 197 020 |
| 3 | $(CH_3)_2NC_6H_{12}$ | 1.9 E-4 | 185 150 |
| 4 | PEG-$b$-PCF$_3$ | 1.4 E-4 | 197 200 |
| 5 | PCF$_3$ | 6.0 E-3 | 197 990 |
| 6 | PEG-OCOMe | 6.9 E-4 | 204 880 |

[a]See supplementary information for polymerization results (Supplementary Figs. 3–8)

core of the micelle; thus, the total volume of reaction is estimated to be roughly ~0.25 mL (when 430 mg of BCP are used). This volume is 400 times smaller than when the polymerization is performed in an organic solvent (100 mL), resulting in an amine concentration 400 times higher in the micelles (assuming it does not leave the micelle). Based on this assumption a prediction for the activity of the catalyst in toluene was determined to be TO ~ 1860 h$^{-1}$ which is in the same order of magnitude as the activity measured in micelles. The inhibition effect of the block copolymer on the catalyst also appears to be surprisingly large in toluene. Therefore, the inhibition constant of each block of the amphiphilic polymer was tested (PEG and PCF$_3$). The homopolymer PCF$_3$ was measured to only exhibit a very small effect on the catalyst activity ($K_{eq} = 6.0$ E-3). The PEG homopolymer showed a higher level of inhibition ($K_{eq} = 6.9$ E-4) which is consistent with a recent report from Chen et al.[70] However, this inhibition remains lower than the one of the BCP. This observation motivated us to investigate the conformation of these polymers in toluene by DLS. While the PCF$_3$ homopolymer appears soluble in toluene only at elevated temperature (>45 °C), the PEG homopolymer forms multiple nanometer size aggregates in toluene at elevated temperature (Supplementary Fig. 12a, b). Finally, the block copolymer PEG-$b$-PCF$_3$ appeared to be perfectly soluble at elevated temperature (no aggregate) (Supplementary Fig. 12c). We postulated from this result that the lower inhibition of the PEG homopolymer in comparison to the block copolymer can be rationalized by the aggregation of the PEG in

toluene at elevated temperature resulting in less PEG capable of inhibition. Overall PEG was found to be one of the main sources of inhibition of the catalyst during the polymerization in micelles.

**Variation in the block copolymer and micelle formation conditions.** The use of different additives was explored as a means to increase the catalyst activity (Table 4). DMF was used in place of THF for the formation of micelles. First, the change of organic solvent did not hamper the catalyst encapsulation, as homogeneous micelles with a slightly larger size (31 nm) were formed. The polymerization with these micelles did not result in any significant change in activity (870 h$^{-1}$, Table 4, entry 1) compared to the activity measured with the micelles made in THF. This constant activity implies first that DMF did not inhibit the catalyst and second that water is unlikely the culprit for the inhibition. Indeed the slightly bigger micelles and the difference in polarity between DMF and THF have likely resulted in micelles with different amounts of water. We tested the influence of pH on our micelle system by decreasing pH of the micelle solution to about 3 with phosphoric acid. The acidified micelles provided a TO frequency of 990 h$^{-1}$ after half an hour (Table 4, entry 2) and 1120 h$^{-1}$ after 1 h (Table 4, entry 3). This enhancement in reactivity with acidified micelles was rationalized as the acid scavenging the labile ligand coordinated to the metal center (tertiary amine) resulting in an increase of the active catalyst concentration. The activity after 0.5 h is slightly lower than when the reaction was run for an hour potentially suggesting a slow scavenging of the amine by the acid. This could be expected as dimethyl hexamine has a greasy chain and has therefore more affinity with the core of the micelle than with water. These results in acidified micelles at different reaction times demonstrate the good protection of the catalyst in the micelles. Additionally the small enhancement of TO.h$^{-1}$ obtained by performing the polymerization at low pH further confirms that amine inhibition is not the main culprit for the low activity observed. Finally polymerization in PEG-$b$-PS BCP micelles was performed (Table 4, entry 4). The activity observed with this non-coordinating apolar block was very similar to the one observed with the PEG-$b$-PCF$_3$ confirming that coordination to the ester group (of the acrylate backbone) or the water are not the main culprits for the lower activity in micelles.

**Extension to another catalyst.** The encapsulation strategy was applied to another palladium phosphinosulfonate catalyst L$_2$Pd-DMSO. This catalyst is of interest because it was shown previously to exhibit very high activity in toluene (774,000 h$^{-1}$, Table 1, entry 5)[60]. Note that L$_2$Pd-DMSO does not require the use of a fatty alkyl ligand for the formation of homogeneous micelles. The polymerization in micelles with L$_2$Pd-DMSO resulted in activity of up to 2400 TO h$^{-1}$. This activity is lower than the activity of the same catalyst in organic solvent but is twice higher than L$_1$Pd-NR$_3$ under otherwise identical reaction conditions. In fact, the activity of the catalyst was so high that we noticed some coagulation in the reactor and a broader particle size distribution of the latex. This colloidal instability was attributed to the insufficient stabilization of the PE particle by the block copolymer. The coagulation was suppressed by lowering the catalyst loading (Table 5, entry 1) and/or by increasing the BCP loading (Table 5, entry 4). The decrease in catalyst loading induced a small decrease in activity (1760 h$^{-1}$) while the increase of BCP loading maintained a high TO frequency of 2130 h$^{-1}$. Despite the absence of coagulate during the polymerization, the polyethylene particle size distribution of the latex formed with L$_2$Pd-DMSO remained broad. We attributed this to an inhomogeneous encapsulation of the catalyst resulting in the formation of

### Table 4 Variation in reaction conditions

| Entry | Reaction changes | Catalyst loading [μmol] | Yield PE [g] | TO.h$^{-1a}$ | PS$^b$ (before) [nm] | PS$^b$ (after) [nm] |
|---|---|---|---|---|---|---|
| 1 | Micelles formed in DMF With PEG-b-PCF$_3$ | 15 | 0.36 | 870 | 30 | 115 |
| 2 | Micelles formed in THF$^{c,d}$ | 14 | 0.19 | 990 | 24 | 80 |
| 3 | Micelles formed in THF With PEG-b-PCF$_3$/pH-3$^c$ | 17 | 0.52 | 1120 | 28 | 163 |
| 4 | Micelles formed in THF With PEG-b-PS | 16 | 0.49 | 1120 | 27 | 67 |

*Reaction conditions*: catalyst: L$_1$Pd-NR$_3$, 85 °C, 40 bar of constant ethylene pressure, 1 h, total volume of solvent: 100 mL
$^a$Mol of ethylene consumed per mol of Pd per hour
$^b$Average particles size by volume before and after ethylene polymerization determined by DLS (See Supplementary Table 4)
$^c$pH adjusted with phosphoric acid (10$^{-3}$ mol L$^{-1}$)
$^d$Polymerization time is 0.5 h

### Table 5 L$_2$Pd-DMSO catalyzed polymerizations of ethylene

| Entry | Catalyst loading [μmol] | Yield PE [g] | TO.h$^{-1\,a}$ | $M_n$ (by GPC) [g mol$^{-1}$] | Đ | PS$^b$ (before) [nm] | PS$^b$ (after) [nm] |
|---|---|---|---|---|---|---|---|
| 1 | 8 | 0.40 | 1760 | 17,710 | 2.3 | 27 | 66 |
| 2 | 16 | 1.1$^d$ | 2430 | 22,810 | 1.8 | 26 | 112 |
| 3$^c$ | 9 | 0.48 | 1830 | 12,650 | 2.1 | 26 | 83 |
| 4$^e$ | 16 | 0.95$^f$ | 2180 | 22,920 | 1.5 | 27 | 103 |

*Reaction conditions*: 85 °C, 430 mg PEG-*b*-PCF$_3$, 40 bar of constant ethylene pressure, 1 h reaction time, total volume of solvent: 100 mL
$^a$Mol of ethylene consumed per mol of Pd per hour
$^b$Average particles size by volume before and after ethylene polymerization determined by DLS (See Supplementary Table 5)
$^c$Addition of N(Me)$_2$C$_6$H$_{13}$ (1 equiv.) in THF
$^d$Quantity of PE found dispersed in water an additional 330 mg of coagulated PE was collected by filtration after polymerization
$^e$Quantity of BCP and THF used to form micelle adjusted to same ratio as entry 1 (764 mg PEG-b-PCF$_3$ and 8.8 mL THF)
$^f$Quantity of PE found dispersed in water an additional 45 mg of coagulated PE was collected by filtration after polymerization. TEM images available in SI (Supplementary Fig. 13, images c and d)

micelles with different catalyst concentrations. This could be avoided by the addition of 1 equivalent of dimethyl hexyl amine labile to the organic solution of L$_2$Pd-DMSO and block copolymer to yield an in situ made L$_2$Pd-NR$_3$. This addition resulted, however, in a small decrease in activity (1830 h$^{-1}$ Table 5, entry 3). Overall the moderate activity of the DMSO catalyst and the minor difference in activity upon addition of NR$_3$ confirm that the inhibition is not exclusively caused by the labile ligand.

## Discussion

We have developed an alternative strategy to perform the catalytic polymerization of ethylene in aqueous media. Our approach consists of encapsulating a water-insoluble catalyst precursor into polymer-based micelles dispersed in water. Block copolymers with the composition PEG-*b*-PCF$_3$ and PEG-*b*-PS were shown to form thermodynamically stable micelles that could be loaded homogeneously with a catalyst precursor. The performances of the encapsulated catalysts studied here were found to be significantly higher than those obtained using traditional mini-emulsion strategy with the same catalyst. The good protection of the catalyst in micelles resulted in an enhanced stability of the catalyst in dispersion. The catalyst activity in micelles remained, however, multiple orders of magnitude lower than in organic solvent. This difference in activity was rationalized by the presumably lower solubility of the monomer in micelles, the relatively high concentration of amine in the micelle and the coordination of the PEG block. Finally the use of acid, of BCP with different chemistry and more active catalyst resulted in an enhancement in activity. The intrinsic relationship between catalyst activity and block copolymer microstructure motivates the development of next generation block copolymers to further enhance the productivity of the catalyst as well as the investigation of the location of the catalyst during the polymerization.

## Methods

All ethylene polymerizations were carried out in mechanically stirred (1000 rpm), 100 mL stainless steel high-pressure reactor equipped with a heating and cooling jacket and thermocouple. The ethylene pressure was kept constant throughout the polymerizations. Before each polymerization run, the reactor was purged under vacuum at 85 °C before to be backfilled with argon. Three quick more cycles of vacuum-and-backfilling with argon were repeated before the transfer of reagents in the reactor.

**Polymerization via micellization**. A specific amount of BCP was dissolved in tetrahydrofuran (5 mL) into a 250 mL Schlenk flask before adding the pre-catalyst. Water (95 mL) was added dropwise to the above solution at 35–40 °C over 15–20 min. This solution was filtered and transferred by cannula to the reactor at 85 °C before being pressurized at the desired pressure of ethylene and left to react for the desired time.

Specific weights of reaction solution were dried at 120 °C overnight. TO were calculated via gravimetric analysis using the weight difference of dry samples before and after ethylene polymerization.

Additional experimental procedures in SI (Supplementary Method).

**Data availability**. The authors declare that the main data supporting the findings of this study are available within the article and its Supplementary Information files. Extra data are available from the corresponding author upon request.

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

## Acknowledgements

This work was supported by the Dow Chemical Company through Grant RPS 226772 AA. The authors would like to thank Andrew Hughes of the Dow Chemical Company for helpful discussions, Mikaela Dressendorfer for her help on some block copolymers synthesis, the Frederick Seitz Materials Research Laboratory for facilities and instrumentation and Dr. Schroeder's group for TEM images (Drs. Bo Li and Songsong Li).

## Author contributions

All the authors participated in the design of the experiments and the preparation of the manuscript. C.B.-J. and M.R. performed the experiments.

## Additional information

**Competing interests:** A United States Provisional Patent relating to this work was filed with the USPTO on which M.R., J.S.K., R.E. and D.G. are named inventors. All remaining authors declare no competing financial interests.

