## [Peer Review File · Nature Communications]

Reviewers' comments:

Reviewer #1 (Remarks to the Author):

This paper presents a novel process to perform the catalytic polymerisation of ethylene in emulsion, using amphiphilic block copolymers as a means to protect and isolate the catalyst in stable micellar nanocontainers. In my opinion, several features of this paper are remarkable. First, the use of block copolymers solves the problems associated with mini/microemulsions strategies (organic solvent, large amount of surfactant, low colloidal stability). Thus, if the catalytic polymerisation of ethylene is to become commercial, then the strategy proposed here is likely to be employed. Second, previous attempts of ethylene emulsion polymerisation were plagued by a decrease of activity in comparison to polymerisations conducted in organic medium. Although it is also the case here, the authors provide for the first time a reasonable explanation for it.

The paper has also several shortcomings. From an industrial point of view, the authors only managed to prepare 300 to 600 mg of polyethylene in 250 to 750 mg copolymer micelles. Thus, the method suffers from the low capacity of the micelles to grow and stabilize polyethylene. The catalyst choice, Pd phosphine sulfonate, is also surprising. Such catalysts have been developed to copolymerise ethylene with polar monomers. Why not use a Ni-based catalyst for ethylene homopolymerisation? Usually Ni catalysts are less expensive and more active. With Pd phosphine sulfonates, examples of copolymerisation would have been expected.

The kinetic/mechanistic analysis is, in my opinion, highly debatable (if not wrong). First, the authors claim that their experiments are not under mass-transfer limitation (page 12, line 207). Yet, TOF is independent of catalyst concentration (Table 3) which usually is a good indication that mass transfer is a problem. There are no catalytic reactions which are zero order with respect to catalyst concentration. It is also indicated (page 114 line 232) that the reaction is zero order with respect to ethylene pressure above 10 bar pressure. I think I have read most (if not all) of the papers of Nozaki, Mecking, Jordan, Claverie and Asua related to ethylene polymerisation with Pd phosphine sulfonates, and I find this statement contradicts what transpires from their studies. Could poor mixing and slow gas-liquid mass transfer responsible for this apparent zero-order with respect to monomer? Note that the raw data should be included (Figure S8 in SI does not give TOF vs ethylene pressure).

According to the authors, inhibition is due to the high concentration of the amine ligand inside the nanoreactor. What about ethylene concentration? It should also be larger in the hydrophobic environment than in water. I would think that the increase in amine concentration is balanced by an increase in ethylene concentration. I was surprised to see that TO remains stable then decreases with time (Figure 5). If inhibition is solely due to the amine concentration, the activity should increase as the size of the nanoreactor increases and the amine concentration decreases with time. Why is inhibition observed for the DMSO catalyst (table 6)? Surely, DMSO is not confined to the inside of the micelle, but it is partitioned with water.

The kinetics analysis is also questionable. Species 2 and 1 are NOT in equilibrium. Using the usual Michealis Menten approach, a steady state approximation should be written for species 2, which leads to

EQUATION ADDED IN SUPPLEMENTARY MATERIALS

Where k_1 and k_{-1} are the kinetic constants for the preequilibrium, Thus equation 1 is generally invalid, unless k_p is small compared to k_{-1} . Furthermore, the passage from equation 2 to 3 is invalid. TOF and TONs are calculated using the TOTAL catalyst concentration. But in equation 3, the TOF was obtained by using $[1]$ in equation 2 instead of the total catalyst concentration. How were the M_n of PE prepared in emulsion determined? Are those the M_n of the mixture of block copolymer and PE?

EQUATION 1 SHOULD BE:

Reviewer #2 (Remarks to the Author):

The paper submitted by Boucher-Jacobs et al is an original and interesting contribution to the the problem of emulsion polymerization of ethylene in aqueous media. As an alternative to the existing procedures, such as mini-emulsion techniques and Ni water-soluble catalytic systems, the present authors have examined the “nano-reactor concept” for the catalytic emulsion polymerization of ethylene in water. The originality of this concept is the fact that a Pd type catalyst is encapsulated in the micellar core of an amphiphilic block copolymer.

The background of this type of studies is clearly outlined in the introduction of the manuscript. All experimental and characterization techniques are perfectly described in the main paper and supporting informations. The repetition of the experiments would therefore be possible.

In a first step of their study , 3 types of amphiphilic diblock copolymers with a constant PEG sequence were prepared , the hydrophobic block being PS; PCF3 and PEHMA respectively. The corresponding micellar systems, including those comprising the encapsulated Pd catalyst, were obtained by selective precipitation with water. After the particle size determination of the micellar dispersions, the emulsion polymerization experiments were carried out exclusively by using the fluorinated copolymer PEG-b-CF3 for the encapsulation of the catalyst.

It could cleanly be demonstrated by the authors that the encapsulated catalyst keeps its efficiency for the polymerization of ethylene in aqueous media and that on the average one micelle generates one PE nanoparticle. The concept of block copolymer micelles acting as nano- reactor has thus been validated. However, with respect to polymerizations performed in organic solvents it appeared an important efficiency decrease for these water-based systems. Different parameters, such as residual THF and/or water, complex formation between catalyst and PEG could be identified by the authors that may lead to an inhibition effect and thus to the efficiency decrease of the catalyst.

At this stage of the authors' investigations, some additional suggestions might have to be taken into account for the continuation of this study'

1) Selection of the nano-reactor block-copolymer

One has to keep in mind that the block copolymer has in fact a double role, that of encapsulant of the catalyst and that of steric stabilizer of the PE

nanoparticles to prevent their agglomeration. This would require that the copolymer (and the catalyst) may be able to migrate to particle surface in the emulsion polymerization step. This rises further the question if the catalyst is molecularly solubilized in the micellar core or if it is present due to its solubility parameter in the *core-shell interphase* of the micelles.

2) *Micelle preparation and their morphology*

Polyethylene block copolymers such as PE-b- PEO, easily accessible by hydrogenation of polybutadiene-b-PEO, may be suggested as the catalyst encapsulant. Moreover, in general practice a micellar dispersion is usually purified by dialysis In order to eliminate the common solvent ; such as THF, of the two blocks.

Various characterization techniques, such as SANS, cryo-TEM; tomography etc are presently available for a detailed analysis of micellar morphologies.

3) *Surface analysis of the PE nanoparticles*

This type of analysis is suggested in order to check if the PE chains grow inside or on surface of the copolymer micelles

CONCLUSION: In conclusion, this manuscript is recommended for its publication in NATURE COM. The results are significant and described in detail. The concept of catalyst loaded block copolymer micelles, in spite of some limitations, opens interesting perspectives for the emulsion polymerization of olefins and for the development of structured nanoparticles

Reviewer #3 (Remarks to the Author):

- What are the major claims of the paper? Are the claims novel? If not, please identify the major papers that compromise novelty

This study describes the development of block copolymer amphiphiles that form aqueous micelles, supporting Pd-catalyzed ethylene polymerization to form PE latexes.

A strength of this study is that it appears to constitute the first examples of ethylene polymerization in a normal aqueous emulsion rather than on-water or miniemulsion. A weakness of this study, in this reviewer's opinion, was the subtle distinction between the PE dispersions formed here versus any of a number of prior studies on metal-catalyzed ethylene polymerization in water and/or in miniemulsion. In the latter cases, these have primarily involved base (Ni) rather than precious (Pd) metal catalysts and are still competent to generate PE dispersions with good uniformity.

Absent a clearly stated strategic advantage of ethylene polymerization in 'normal' versus mini emulsion, the authors' main criticism of prior work appears to be that polymerization rates were too low. However, the absolute catalyst activities in the present work are also quite modest. The claim of 'multiple orders of magnitude higher' (line 336) activity for their aqueous polymerizations doesn't seem to comport with literature data for other emulsion ethylene polymerizations. The highest turnover frequency in this study (2400 h⁻¹) is inferior to values reported by Mecking (18500 h⁻¹, *Macromolecules* 2007, 40, 421; 45500 h⁻¹, *JACS* 2017, 139, 13786) or Claverie (>100000 h⁻¹, *Macromolecules* 2001, 34, 2022), for example. If instead the comparison was meant to be versus their control reaction (line 106) using the Drent catalyst in miniemulsion, this may simply indicate the chosen control reaction poorly represents a typical miniemulsion polymerization using Ni catalysts.

- Will the paper be of interest to others in the field? Will the paper influence thinking in the field?

I do believe the emulsifiers identified in this study that enable polymerization in micelles while maintaining catalyst stability will be of interest to those working in the area of alkene polymerization. This compliments the general approaches that have to date relied heavily on surfactants, such as SDS.

The study appears to have been thoroughly conducted, particularly with respect to establishing the size and distribution of micelles and PE particles that are formed through DLS measurements. A systematic study on factors influencing catalyst activity during polymerization also helps to understand what may or may not significantly inhibit the catalyst under aqueous emulsion conditions.

Overall, I think the novelty of the block copolymer emulsifiers is compelling and could be broadly useful in emulsion polymerizations. This consideration, in addition to the mechanistic insights into catalyst inhibition in emulsion, outweigh the modest catalyst activity and similarity of the resultant PE dispersions versus past work in this area using miniemulsions.

- Are the claims convincing? If not, what further evidence is needed?

While water was ruled out as a significant contributor to catalyst inhibition in these reactions, these conclusions seem to be predicated on the action of water specifically as a dative ligand. Water could also be inhibitory through its chemical reactivity, such as through protodepalladation of the propagating species. The resultant [Pd]-OH would be deactivated towards initiation of a new polymer chain through slower oxypalladation (Wacker-type reaction). Due to the presence of residual PEG in the PE products, any hydroxyl PE chain end resonances would likely be obscured by NMR were this side reaction occurring. A simple test to probe for this alternative role of water in

throttling rates would be to conduct a polymerization in heavy water and observing $^{13}\text{C}/^2\text{H}$ coupling at the methyl PE chain ends by ^{13}C NMR, which is more sensitive than ^2H NMR (the fact that low pH didn't exacerbate this potential issue is moot, since hydronium ion would be far less likely to enter the micelle).

- Are there other experiments that would strengthen the paper further? How much would they improve it, and how difficult are they likely to be?

The authors have already provided a series of experiments that inform the different inhibitory efficacy of solvents, ancillary ligands, and emulsifiers. The above experiment to probe the potential inhibitory role of water beyond acting as a simple dative ligand would further strengthen their counter-intuitive conclusion that water doesn't inhibit polymerization rates to any significant extent.

- Are the claims appropriately discussed in the context of previous literature?

Yes

- If the manuscript is unacceptable in its present form, does the study seem sufficiently promising that the authors should be encouraged to consider a resubmission in the future?

Yes

Additional comment: The key catalyst activities reported in Table 1, entries 7-8 and on line 308 are incorrect and too high for the PE mass yields reported. There is either an error in the calculation of these values or the respective parameters in Table 1 have a typo.

Response to Reviewers' comments:

Reviewer #1 (Remarks to the Author):

This paper presents a novel process to perform the catalytic polymerisation of ethylene in emulsion, using amphiphilic block copolymers as a means to protect and isolate the catalyst in stable micellar nanocontainers. In my opinion, several features of this paper are remarkable. First, the use of block copolymers solves the problems associated with mini/microemulsions strategies (organic solvent, large amount of surfactant, low colloidal stability). Thus, if the catalytic polymerisation of ethylene is to become commercial, then the strategy proposed here is likely to be employed. Second, previous attempts of ethylene emulsion polymerisation were plagued by a decrease of activity in comparison to polymerisations conducted in organic medium. Although it is also the case here, the authors provide for the first time a reasonable explanation for it.

The paper has also several shortcomings. From an industrial point of view, the authors only managed to prepare 300 to 600 mg of polyethylene in 250 to 750 mg copolymer micelles. Thus, the method suffers from the low capacity of the micelles to grow and stabilize polyethylene. The catalyst choice, Pd phosphine sulfonate, is also surprising. Such catalysts have been developed to copolymerise ethylene with polar monomers. Why not use a Ni-based catalyst for ethylene homopolymerisation? Usually Ni catalysts are less expensive and more active. With Pd phosphine sulfonates, examples of copolymerisation would have been expected.

Answer: We picked this palladium system for its very high activity in organic solvent (L2Pd catalyst has one of the highest activities reported for a neutral late transition catalyst) and poor activity in water. This low activity in aqueous media made it easier for us to highlight the benefit of our encapsulation strategy. This paper is our first paper demonstrating the feasibility of a micellar encapsulation strategy but surely we intend to apply it next to a more industrially relevant nickel catalyst.

The kinetic/mechanistic analysis is, in my opinion, highly debatable (if not wrong). First, the authors claim that their experiments are not under mass-transfer limitation (page 12, line 207). Yet, TOF is independent of catalyst concentration (Table 3) which usually is a good indication that mass transfer is a problem. There are no catalytic reactions which are zero order with respect to catalyst concentration. It is also indicated (page 114 line 232) that the reaction is zero order with respect to ethylene pressure above 10 bar pressure. I think I have read most (if not all) of the papers of Nozaki, Mecking, Jordan, Claverie and Asua related to ethylene polymerisation with Pd phosphine sulfonates, and I find this statement contradicts what transpires from their studies.

Answer: Late transition metal catalysts have often been demonstrated to show saturation kinetics. The very first palladium catalyst reported for ethylene polymerization was reported to be zero order in ethylene. (Brookhart, *Macromolecules*, 2001, 34, 1140). Mecking reported in 2009 that the exact catalyst we use here exhibits saturation kinetics above 5 atm when a highly labile dmso is used for the catalyst precursor (*JACS* 2009, 422). The same group followed up with another paper (*Organometallics*, 2012, 3128) where they compared the activity of different catalyst precursor all coordinated to highly labile ligands and showed that past 5 atm they all had the same activity (*Organometallics*, 2012, 3128). Therefore the saturation kinetics we observed with NR_3 is thus in-line with these previous reports.

Could poor mixing and slow gas-liquid mass transfer responsible for this apparent zero-order with respect to monomer? Note that the raw data should be included (Figure S8 in SI does not give TOF vs ethylene pressure).

Answer: The raw data was included in the supporting information. We made only 2.6 g of PE during these experiments (see updated Figure S16). We have, under the exact same conditions (temperature and pressure) but with higher concentrations of catalyst or with L2PdDMSO, made larger amounts of PE (up to 2 g in 15min). This demonstrates that the saturation kinetics is not due to mass transfer limited diffusion of ethylene gas into toluene.

According to the authors, inhibition is due to the high concentration of the amine ligand inside the nanoreactor. What about ethylene concentration? It should also be larger in the hydrophobic environment than in water. I would think that the increase in amine concentration is balanced by an increase in ethylene concentration. I was surprised to see that TO remains stable then decreases with time (Figure 5). If inhibition is solely due to the amine concentration, the activity should increase as the size of the nanoreactor increases and the amine concentration decreases with time. Why is inhibition observed for the DMSO catalyst (table 6)? Surely, DMSO is not confined to the inside of the micelle, but it is partitioned with water.

Answer: We apologize if our manuscript was lacking clarity. We state that the high amine concentration could be the culprit for the low activity but the coordination to the PEG unit of the block copolymer appears to play an even bigger role. The enhanced activity observed at lower pH shows that the amine did indeed play a role but is not the sole culprit for the low activity. As the reviewer pointed out, the fact that the DMSO complex does not yield higher activity confirms that the amine is not the sole culprit.

The kinetics analysis is also questionable. Species 2 and 1 are NOT in equilibrium. Using the usual Michealis Menten approach, a steady state approximation should be written for species 2, which leads to

EQUATION 1 SHOULD BE:

$$k_1 [1][C_2H_4] = k_{-1} [2][L] + k_p [2]$$

Where k_1 and k_{-1} are the kinetic constants for the preequilibrium, Thus equation 1 is generally invalid, unless k_p is small compared to $k_{-1} L$. Furthermore, the passage from equation 2 to 3 is invalid. TOF and TONs are calculated using the TOTAL catalyst concentration. But in equation 3, the TOF was obtained by using [1] in equation 2 instead of the total catalyst concentration.

Answer. Thank you for the suggestion. We, indeed presume a very fast pre-equilibrium. This assumption is based on the fact that the olefin complex has never been observed for this catalyst (Organometallics, 2012, 3128, and JACS 209, 422). Furthermore, the kinetic model we used here has previously been used for this catalyst (Macromolecules, 2009, 6953) but also for other late transition metal based olefin polymerization catalyst exhibiting the same saturation kinetics (JACS, 204, 5827).

Below is the entire derivation that lead us to equation 3:

$$R_p = k_p[2]$$

$$K_{eq} = \frac{[2][L]}{[1][C_2H_4]}$$

$$[Pd]_o = [2] + [1]$$

$$K_{eq} = \frac{[2][L]}{([Pd]_o - [2]) * [C_2H_4]}$$

$$[C_2H_4] * K_{eq} * [Pd]_o = [2] * ([L] + K_{eq}[C_2H_4])$$

$$[2] = \frac{[Pd]_o}{\frac{[L]}{K_{eq}[C_2H_4]} + 1}$$

$$\frac{R_p}{[Pd]_o} = \frac{k_p}{\frac{[L]}{K_{eq}[C_2H_4]} + 1}$$

$$TOF = \frac{R_p}{[Pd]_o}$$

$$\frac{1}{TOF} = \frac{\frac{[L]}{K_{eq}[C_2H_4]} + 1}{k_p}$$

$$\frac{1}{TOF} = \frac{[L]}{K_{eq}k_p[C_2H_4]} + \frac{1}{k_p}$$

How were the Mn of PE prepared in emulsion determined? Are those the Mn of the mixture of block copolymer and PE?

Answer

The polyethylene is isolated by drying of the latex. Once a solid is isolated, this solid is washed with large amount of warm THF to remove the block copolymer (Shaker set at 70C), therefore the MW determined by GPC are of the pure PE. This information has now been added to the document

Reviewer #2 (Remarks to the Author):

The paper submitted by Boucher-Jacobs et al is an original and interesting contribution to the the problem of emulsion polymerization of ethylene in aqueous media. As an alternative to the existing procedures, such as miniemulsion techniques and Ni water-soluble catalytic systems, the present authors have examined the “nano-reactor concept” for the catalytic emulsion polymerization of ethylene in water. The originality of this concept is the fact that a Pd type catalyst is encapsulated in the m the micellar core of an amphiphilic block copolymer.

The background of this type of studies is clearly outlined in the introduction of the manuscript. All experimental and characterization techniques are perfectly described in the main paper and supporting informations. The repetition of the experiments would therefore be possible. In a first step of their study, 3 types of amphiphilic diblock copolymers with a constant PEG sequence were prepared, the hydrophobic block being PS; PCF3 and PEHMA respectively. The corresponding micellar systems, including those comprising the encapsulated Pd catalyst, were obtained by selective precipitation with water. After the particle size determination of the micellar dispersions, the emulsion polymerization experiments were carried out exclusively by using the fluorinated copolymer PEG-b-CF3 for the encapsulation of the catalyst.

It could clearly be demonstrated by the authors that the encapsulated catalyst keeps its efficiency for the polymerization of ethylene in aqueous media and that on the average one micelle generates one PE nanoparticle. The concept of block copolymer micelles acting as nano-reactor has thus been validated. However, with respect to polymerizations performed in organic solvents it appeared an important efficiency decrease for these water-based systems. Different parameters, such as residual THF and/or water, complex formation between catalyst and PEG could be identified by the authors that may lead to an inhibition effect and thus to the efficiency decrease of the catalyst. At this stage of the authors' investigations, some additional suggestions might have to be taken into account for the continuation of this study'

Answer. Thank you very much for some great suggestions. While we will address every point raised, this additional work will be part of later publications.

1) Selection of the nano-reactor block-copolymer. One has to keep in mind that the block copolymer has in fact a double role, that of encapsulant of the catalyst and that of steric stabilizer of the PE nanoparticles to prevent their agglomeration. This would require that the copolymer (and the catalyst) may be able to migrate to particle surface in the emulsion polymerization step. This rises further the question if the catalyst is molecularly solubilized in the micellar core or if it is present due to its solubility parameter in the core-shell interphase of the micelles.

Answer. This is a very good point. While we are very confident that the catalyst precursor is encapsulated in the micelle initially. We do not know exactly where the catalyst is during the polymerization. We presume, as suggested by the reviewer that the catalyst is at the interface between the semi-crystalline PE and the hydrophilic PEG but we do not have any physical evidence supporting that.

2) Micelle preparation and their morphology Polyethylene block copolymers such as PE-b- PEO, easily accessible by hydrogenation of polybutadiene-b-PEO, may be suggested as the catalyst encapsulant. Moreover, in general practice a micellar dispersion is usually purified by dialysis in order to eliminate the common solvent; such as THF, of the two blocks. Various characterization techniques, such as SANS, cryo-TEM; tomography etc are presently available for a detailed analysis of micellar morphologies.

Answer. Thanks for the suggestion. We are currently investigating other block copolymers but we are trying to move away from the PEO block as our study suggests it causes the inhibition. We did not analyze our micelles beyond DLS. We were using composition that were already demonstrated to form spherical structure micelles (Macromol. Chem. Phys., 1996, 197, 3697). The good agreement between

the experimental diameters of the micelles we measured and the previous literature gave us enough confidence that we had formed the desired spherical micelles.

We initially removed the THF present in the solution by simply bubbling N₂ into the flask. After 1-hour of continuous bubbling, the micelle solution did not have any THF smell. The polymerization with this THF depleted micelle did not result in any improvement in activity and therefore we stopped removing the solvent. We did not include this experiment in the manuscript because it was tedious and we did not quantify precisely the level of residual THF in the latex

3) Surface analysis of the PE nanoparticles This type of analysis is suggested in order to check if the PE chains grow inside or on surface of the copolymer micelles

Answer. This is a great suggestion and we will surely include it our list of future experiments.

CONCLUSION: In conclusion, this manuscript is recommended for its publication in NATURE COM. The results are significant and described in detail. The concept of catalyst loaded block copolymer micelles, in spite of some limitations, opens interesting perspectives for the emulsion polymerization of olefins and for the development of structured nanoparticles

Reviewer #3 (Remarks to the Author):

- What are the major claims of the paper? Are the claims novel? If not, please identify the major papers that compromise novelty

This study describes the development of block copolymer amphiphiles that form aqueous micelles, supporting Pd-catalyzed ethylene polymerization to form PE latexes.

A strength of this study is that it appears to constitute the first examples of ethylene polymerization in a normal aqueous emulsion rather than on-water or miniemulsion. A weakness of this study, in this reviewer's opinion, was the subtle distinction between the PE dispersions formed here versus any of a number of prior studies on metal-catalyzed ethylene polymerization in water and/or in miniemulsion. In the latter cases, these have primarily involved base (Ni) rather than precious (Pd) metal catalysts and are still competent to generate PE dispersions with good uniformity.

Absent a clearly stated strategic advantage of ethylene polymerization in 'normal' versus mini emulsion, the authors' main criticism of prior work appears to be that polymerization rates were too low. However, the absolute catalyst activities in the present work are also quite modest. The claim of 'multiple orders of magnitude higher' (line 336) activity for their aqueous polymerizations doesn't seem to comport with literature data for other emulsion ethylene polymerizations. The highest turnover frequency in this study (2400 h⁻¹) is inferior to values reported by Mecking (18500 h⁻¹, *Macromolecules* 2007, 40, 421; 45500 h⁻¹, *JACS* 2017, 139, 13786) or Claverie (>100000 h⁻¹, *Macromolecules* 2001, 34, 2022), for example. If instead the comparison was meant to be versus their control reaction (line 106) using the Drent catalyst in miniemulsion, this may simply indicate the chosen control reaction poorly represents a typical miniemulsion polymerization using Ni catalysts.

Answer. Thank you for the suggestion. This statement was intended to the phosphinosulfonate palladium system and not to late transition metal catalyst in general. We adjusted the statement made in the manuscript to represent more precisely the literature. After careful review of all the papers suggested by the reviewer and others we confirmed that activities in organic solvent remain systematically higher than in aqueous medium (miniemulsion or pure water). This difference in activity was the motivation for our project. With that said the activity we obtained in water remained also significantly below the one observed in organic solvent, motivating for more research.

Table 1: Overview of activities reported in the literature between organic and aqueous medium

REF	Activity org solvent	Activity w/ water	Units	Reaction medium
JACS, 2017, 139, 13786 Mecking	200 000	46 000	mol(C ₂ H ₄)/mol(cat).h	water
Macromolecules, 2001, 34, 2022, Saudemont	>2 000 000	83 000	g(PE)/g(Ni).h	miniemulsion
Macromolecules, 2009, 42, 3669, Mecking	138 000	6 000	mol(C ₂ H ₄)/mol(cat).h	water
Macromolecules, 2007, 40, 421, Mecking		19000	mol(C ₂ H ₄)/mol(cat).h	miniemulsion
Organometallics, 2007, 26, 1311, Mecking	84 000	3 000 9 000	mol(C ₂ H ₄)/mol(cat).h	H ₂ O H ₂ O/iPrOH
Bauers, Zuideveld, Thomann, Mecking	42 000	5 600	mol(C ₂ H ₄)/mol(cat).h	miniemulsion

- Will the paper be of interest to others in the field? Will the paper influence thinking in the field?

I do believe the emulsifiers identified in this study that enable polymerization in micelles while maintaining catalyst stability will be of interest to those working in the area of alkene polymerization. This compliments the general approaches that have to date relied heavily on surfactants, such as SDS.

The study appears to have been thoroughly conducted, particularly with respect to establishing the size and distribution of micelles and PE particles that are formed through DLS measurements. A systematic study on factors influencing catalyst activity during polymerization also helps to understand what may or may not significantly inhibit the catalyst under aqueous emulsion conditions.

Overall, I think the novelty of the block copolymer emulsifiers is compelling and could be broadly useful in emulsion polymerizations. This consideration, in addition to the mechanistic insights into catalyst inhibition in emulsion, outweigh the modest catalyst activity and similarity of the resultant PE dispersions versus past work in this area using miniemulsions.

- Are the claims convincing? If not, what further evidence is needed?

While water was ruled out as a significant contributor to catalyst inhibition in these reactions, these conclusions seem to be predicated on the action of water specifically as a dative ligand. Water could also be inhibitory through its chemical reactivity, such as through protodepalladation of the propagating species. The resultant [Pd]-OH would be deactivated towards initiation of a new polymer chain through slower oxypalladation (Wacker-type reaction). Due to the presence of residual PEG in the PE products, any hydroxyl PE chain end resonances would likely be obscured by NMR were this side reaction occurring. A simple test to probe for this alternative role of water in throttling rates would be to conduct a polymerization in heavy water and observing $^{13}\text{C}/^2\text{H}$ coupling at the methyl PE chain ends by ^{13}C NMR, which is more sensitive than ^2H NMR (the fact that low pH didn't exacerbate this potential issue is moot, since hydronium ion would be far less likely to enter the micelle).

Answer. Thank you for the great suggestion. It is true that we did not consider the possibility of water reacting with the catalyst. To the best of our knowledge there is no experimental evidence in the literature of a palladium hydroxyl catalyst capable of initiating ethylene polymerization. There are two computational papers from Ziegler that investigate the reaction of late transition based olefin polymerization catalysts with water but the paper is not conclusive regarding the possibility of initiating the polymerization from a Pd-OH (Inorg. Chem. 2005, 7806 and Organometallics, 2005, 2679). In past experimental papers, polymerizations in water with Brookhart diimine palladium catalyst did not yield any hydroxyl terminated polyethylene (Mecking Angewandte 2006, 6044, Mecking Chem. Commun. 2000, 301). Claverie did a very systematic study of the phosphinosulfonate palladium catalyst in presence of water and did not show any evidence of palladium hydroxyl formation, nor protonation of the palladium alkyl bond (Macromolecules, 2009, 42, 6953).

Following Mecking's recent Macromolecules paper (Macromolecules 2016, 49, 8825) where it was reported that a nickel catalyst is more stable in D_2O than in H_2O , we had performed a polymerization in polymeric micelles dispersed in D_2O prior to submitting the paper. We did not measure any difference in activity (TOF = 830 h^{-1} , PS (after) = 124nm). We later realized that catalyst instability was not the culprit for the low activity and thus we did not include this polymerization in the manuscript. Based upon the suggestion by the reviewer we analyzed the ^1H and ^{13}C NMR spectra of the polyethylene made in this deuterated medium and did not observed any evidence for the presence of CH_2D -PE polymer end-groups (Figure 1, sharp singlet at 14ppm corresponding to the saturated methyl end group of the polymer).

Figure 1: ^{13}C NMR spectrum of a polyethylene made in micelles dispersed in D_2O (150MHz, $\text{C}_2\text{D}_2\text{Cl}_4$, 120°C)

Figure 2: ^1H NMR spectrum of a polyethylene made in micelles dispersed in D_2O (600MHz, $\text{C}_2\text{D}_2\text{Cl}_4$, 120°C)

- Are there other experiments that would strengthen the paper further? How much would they improve it, and how difficult are they likely to be?

The authors have already provided a series of experiments that inform the different inhibitory efficacy of solvents, ancillary ligands, and emulsifiers. The above experiment to probe the potential inhibitory role of water beyond acting as a simple dative ligand would further strengthen their counter-intuitive conclusion that water doesn't inhibit polymerization rates to any significant extent.

- Are the claims appropriately discussed in the context of previous literature?

Yes

- If the manuscript is unacceptable in its present form, does the study seem sufficiently promising that the authors should be encouraged to consider a resubmission in the future?

Yes

Additional comment: The key catalyst activities reported in Table 1, entries 7-8 and on line 308 are incorrect and too high for the PE mass yields reported. There is either an error in the calculation of these values or the respective parameters in Table 1 have a typo.

Answer. Thank you for catching our mistake. The time was different for these polymerizations. A corrected footnote was added to the manuscript.

REVIEWERS' COMMENTS:

Reviewer #1 (Remarks to the Author):

Most of my initial comments remain unanswered. I like the concept of using block-copolymer micelles as nanoreactors, but the results are comparable to already published ones (see table below as well as table in rebuttal). From a synthetic point of view, I cannot fathom how the preparation of an HDPE emulsion with a mere 11g/L solid content using an expensive Pd catalyst and a massive amount of a complicated BCP stabilizer is of interest, when more of the same product can be prepared using a cheaper nickel catalyst and a commercial stabilizer (such as SDBS).

See table on next page.

In their rebuttal, the authors indicate that the catalyst was chosen because it has a very low activity in water but a high activity when confined within BCPs. To support their claim, a 'failed' miniemulsion experiment is reported in Table 2 (the experimental conditions for this miniemulsion experiment should be reported). Such a claim is hard to believe, because closely related Pd phosphine sulfonate catalysts have been successfully used for ethylene miniemulsion polymerizations and copolymerizations. For example, in *Chemical Engineering Journal*, 2011, 168, 1319 (entry 12 Table 2), a TOF of 239 10^{-5} hr⁻¹ Pa⁻¹ is reported for an ethylene miniemulsion polymerization performed at a pressure of 35.5 10^5 Pa, which corresponds to a TOF of 8300 hr⁻¹ - far above any of the activities reported in this paper. Of course, experimental conditions are quite different and apples and oranges should not be compared, but, based on these aforementioned data, I do not understand why miniemulsions failed so badly in the hands of the authors. In fact, the authors do not give any good reasons. They claim that BCP process is 'better', but we need to know why. I suspect the authors did not give a 'fair chance' to a more traditional miniemulsion process. The authors have now included a Figure in SI (S16 - unless mistaken, it was not included in the first version of the manuscript) indicating that activity is independent of pressure above 5 bars for the catalyst bound to the amine ligand. This indeed is indicative of a complete displacement of the amine by ethylene. As in my first review, I find this point surprising, as it was clearly not observed by Jordan, Nozaki or Claverie in the past. For example, see entry 11-15 in Table 1 of *Organometallics* 2007, 26 (26), 6624-6635 or Table 2 in *Macromolecules* 2009, 42 (18), 6953-6963. Note that in *Organometallics*, 2012, 3128 (a paper cited in support by the authors), the catalyst is bound to DMSO or to a phosphine oxide. Not surprisingly, such weak ligands are easily displaced by ethylene. The authors should therefore explain why their amine bound to catalyst is so easily displaced by ethylene (could it be the difference in nucleophilicity between a tertiary amine and pyridine/lutidine)? Another point which should be clarified by the authors is whether this point relevant in BCP emulsion. At a same pressure, how much ethylene is dissolved in the BCP/water system compared to toluene?

The authors did not properly address my point on mass transport limitation, most notably in an emulsion process. The argument brought by the authors is very weak (2g of PE in 15 minutes vs 2.6 g in 30 minutes). Considering the very complicated diffusional path of ethylene (from gas phase to water, then from water to various compartments of BCP and finally to the catalyst), the authors must consider whether physical processes, and not only chemical processes, do play a role in their kinetic observations. For example, essential reaction parameters such as catalyst concentration, number of BCP particles and stirring rate should be considered.

The use of an equilibrium constant in equation 2 remains WRONG. The authors' justification (olefin complex not observed) is nonsensical. Spectroscopic studies are performed under conditions which are very different from kinetic experiments. I also note that such claim contradicts the surprising finding (see above) that the catalyst is saturated with ethylene. Surely, if the catalyst was so easily saturated with ethylene, the olefin complex would have been observed or trapped. Even worse, to back-up their (wrong) kinetic model, the authors claim that it was also used in *Macromolecules*, 2009, 6953. It is not the case. A conventional steady-state approximation is used (the term $k^{-1}L$ was present in the kinetic development, see SI). Last, the mathematical derivation of equation 3 in their refutation is also very dubious. What is L? the amine? the added ligand? the sum of both? L is not a constant term (Total L = Bound L + Free L).

Two additional points: How was the competition experiment performed with water? The concentration of water in toluene is miniscule. Table 4 should include units and experimental conditions (which catalyst, pressure and temperature).

Ref	Amount of polyethylene (highest reported)	Amount of organic solvent	Colloidal stability
This paper	11 grams / Liter (entry 2 Table 6)	50 mL THF / Liter	Good
Macromolecules 2003, 36, 6711	104 grams / Liter (entry 8 Table 1)	50 mL toluene + 3 mL hexadecane / Liter	Good
Chemical Engineering Journal, 2011, 168, 1319-1330	38.6 grams / Liter (entry 16 Table 2)	132 mL toluene / Liter	Good
Macromolecules 2001, 34, 2022	24 grams / Liter (entry 13 Table 1)	33 mL toluene + 3 mL hexadecane / Liter	Good

Reviewer #2 (Remarks to the Author):

NATURE COM referee report

NC COMS 17- 28 165 A Revised version

Herewith I confirm that the authors have perfectly answered the questions listed in my first referee report of this manuscript.

This paper, due to its original aspects, may be recommended for its publication in Nature Com even if further experiments would be required to confirm the concept of nano-reactor.
Polymerization of ethylene.

Furthermore, I agree that my reviewer report and my comments may be published with my name

G. RIESS

Reviewer #3 (Remarks to the Author):

With consideration of the author responses to all reviewer comments and the additional experimental data to test the possible role of water in chain transfer/termination, I support publication of this revised manuscript in Nature Comm.

Reviewer #4 (Remarks to the Author):

I was asked to comment on the authors' replies to reviewer 1 regarding the kinetic and mechanistic studies performed. My comments are limited to this section and I did not evaluate the manuscript in total. Reviewer 1 required clarification/correction of several points of the kinetic/mechanistic analysis. In my opinion the authors did not or only unsatisfyingly answer the major part of his questions. Alternatively they apologized for an unclear representation, but did not undertake any changes in the manuscript to clarify their presentation. On another occasion, they misquoted a publication in the rebuttal nearly to its opposite. Although I find the quality of their replies unsatisfactory to unacceptable, I have to side – for the most part – with the authors regarding their kinetic analysis. With one exception (equation (2) in the manuscript, see below), their analysis is correct and suitable for publication and I have to refute the reviewer's criticism.

In more detail:

1. Presumed zero-order in catalyst concentration

The reviewer was surprised by the zero order in catalyst concentration observed in table 3 which he correctly considers unlikely or indicating mass transport problems. The authors did not reply to this.

I suspect that the reviewer misunderstood the authors' representation. The turn-over-frequency (TOF) in table 3 is presented as "mol of ethylene consumed per mol of Pd per hour". Defined as such, the expected linear dependence on Pd concentration should yield a constant TOF, which the authors indeed observed. The reviewer was probably misled by the authors' statement "This constant activity at various [...] catalyst loading suggests that the polymerization is not mass transfer limited." A slightly different expression, e. g. "The constant TOF per Pd at various...", might avoid this misunderstanding.

2. Zero-order concentration in ethylene concentration, possibly indicating mass transfer limitations

The reviewer questioned the possibility of saturation behaviour in Pd catalysed polymerization and raised the question whether the latter might not indicate mass transfer complications.

I concur completely with the authors' reply that there is no inherent reason to exclude saturation behaviour, i. e. an ethylene coordinated Pd complex becoming the catalyst resting state, for these complexes. The authors convincingly refer to the 2009 Mecking paper (ref. 65) where the same catalyst indeed showed saturation of activity above 5 bar ethylene. The fact the authors were able to obtain higher amounts (mass) of PE at the same ethylene pressures using different catalyst or catalyst loadings likewise indicate that mass transfer limitations are not a problem.

3. Inhibition effects

I concur with the authors' replies to the reviewer's comment.

However, this section required more than one reading to follow the authors' arguments. Unfortunately, although the authors apologized for the lacking clarity, they failed to make any changes to the text. A concluding remark that inhibition by PEG is the most likely culprit before the final conclusion, would be helpful. Likewise, the authors report that acidification enhances activity due to removal of amine, but they did not comment that the observed increase is trivial compared to the reduction of activity going from toluene to micelles. The authors did not discuss the

influence on ethylene concentration in the micelles on activity, until the conclusion.

4. Steady-state vs. pre-equilibrium approach

The reviewer remarks that the authors should apply a steady-state approach, comparable to Michaelis-Menten kinetics, rather than a fast pre-equilibrium approach. He states, correctly, that a pre-equilibrium approach is only justified if " k_p is small compared to $k_{-1}L$ ".

The authors state that an "olefin complex has never been observed for this catalyst". This forms the basis for a steady-state approach, but does **not** validate the further assumption of a fast pre-equilibrium, which relies on the relative rates of ligand attack on 2 vs. ethylene insertion. Simply stating that the same approach has been used previously likewise does not provide an answer to the reviewer's criticism. In this context, one should note that the authors cite the 2009 Claverie Macromolecules paper as having previously used a pre-equilibrium approach. I looked up this article and found that Claverie started from the steady-state assumption the reviewer proposes, not from a pre-equilibrium assumption as the authors claim. Even more, Claverie et al state explicitly: "Contrarily to our initial belief, the insertion step is **not** preceded by a fast pre-equilibrium of ethylene coordination: both reactions go at comparable rates".

I believe, however, that the authors' approach of a pre-equilibrium is the better one in this case, for reasons they did not discuss: In the two extremes of the steady-state approach, we can assume that $k_{-1}L \gg k_p$ (i. e. a pre-equilibrium). In this case, the observed rate constant would be given by $k_{obs} = K[C_2H_4]k_p/[L]$. In the other extreme, $k_{-1}L \ll k_p$, we obtain $k_{obs} = k_1[C_2H_4]$, independent from $[L]$ (propagation rate determined by ethylene coordination, the intermediate trapped by immediate insertion). For $k_{-1}L \approx k_p$, we expect a more complicated behaviour with an apparent reaction order in L between 0 and 1. The fact that the authors observed a well-defined reaction order of -1 for the concentration of the inhibiting ligand L , indicates that k_p is indeed small compared to $k_{-1}L$. In this case a pre-equilibrium approach is to be preferred over a steady-state assumption, since it more correctly describes the behaviour at high monomer concentration.

5. Reviewer : "Furthermore, the passage from equation 2 to 3 is invalid. TOF and TONs are calculated using the TOTAL catalyst concentration. But in equation 3, the TOF was obtained by using $[1]$ in equation 2 instead of the total catalyst concentration."

The authors did not reply directly to this criticism but simply provided their derivation of equation (3). As the authors show in their answer, equation 3 is indeed correctly obtained considering that $[Pd]_0 = [1] + [2]$. Otherwise the term " $+ 1/k_p$ " would be missing in equation 3.

However, the reviewer is indeed correct that equation (2) **is wrong in this sequence**. In fact, no equation identical to or a derivative of equation (2) appears in the authors' rebuttal. As the authors show themselves, substitution of $[1]$ by $[Pd]_0 - [2]$ has to take place, **before** this is used to replace $[2]$ in $R_p = k_p[2]$. Equation (2) thus needs to be replaced by the $R_p/Pd_0 = \dots$ equation (or a derivative thereof) from their rebuttal. It would be very useful to include the full derivation in the supporting information.

RESPONSE TO REVIEWERS' COMMENTS:

Reviewer #1 (Remarks to the Author):

Most of my initial comments remain unanswered. I like the concept of using block-copolymer micelles as nanoreactors, but the results are comparable to already published ones (see table below as well as table in rebuttal). From a synthetic point of view, I cannot fathom how the preparation of an HDPE emulsion with a mere 11g/L solid content using an expensive Pd catalyst and a massive amount of a complicated BCP stabilizer is of interest, when more of the same product can be prepared using a cheaper nickel catalyst and a commercial stabilizer (such as SDBS).

Ref	Amount of polyethylene (highest reported)	Amount of organic solvent	Colloidal stability
This paper	11 grams / Liter (entry 2 Table 6)	50 mL THF / Liter	Good
Macromolecules 2003, 36, 6711	104 grams / Liter (entry 8 Table 1)	50 mL toluene + 3 mL hexadecane / Liter	Good
Chemical Engineering Journal, 2011, 168, 1319-1330	38.6 grams / Liter (entry 16 Table 2)	132 mL toluene / Liter	Good
Macromolecules 2001, 34, 2022	24 grams / Liter (entry 13 Table 1)	33 mL toluene + 3 mL hexadecane / Liter	Good

Response: We agree that our approach yields a very expensive polyethylene latex. Our stabilizing agent is complicated to synthesize, the metal we employ is expensive, and the activity of our catalyst remains low. The use of the solid content as a comparison point is not the most adequate, as we could have easily increased our solid content by increasing our micelle concentration. The key parameter for success is the overall activity of the catalyst reported as TOF. A clearer definition of TOF has now been included in the manuscript.

In their rebuttal, the authors indicate that the catalyst was chosen because it has a very low activity in water but a high activity when confined within BCPs. To support their claim, a 'failed' miniemulsion experiment is reported in Table 2 (the experimental conditions for this miniemulsion experiment should be reported). Such a claim is hard to believe, because closely related Pd phosphine sulfonate catalysts have been successfully used for ethylene miniemulsion polymerizations and copolymerizations. For example, in Chemical Engineering Journal, 2011, 168, 1319 (entry 12 Table 2), a TOF of $239 \times 10^{-5} \text{ hr}^{-1} \text{ Pa}^{-1}$ is reported for an ethylene miniemulsion polymerization performed at a pressure of 35.5 MPa, which corresponds to a TOF of 8300 hr^{-1} - far above any of the activities reported in this paper. Of course, experimental conditions are quite different and apples and oranges should not be compared, but, based on these aforementioned data, I do not understand why miniemulsions failed so badly in the hands of the authors. In fact, the authors do not give any good reasons. They claim that BCP process is 'better', but we need to know why. I suspect the authors did not give a 'fair chance' to a more traditional miniemulsion process.

Response: We did not intend to decrease the value of the miniemulsion strategy but rather to highlight that activities thus far in water or in miniemulsion have remained lower than in organic solvent. This observation motivated the development for an alternative strategy. In the main manuscript, we

rephrased the statements regarding the direct comparison of miniemulsion and our technique to ensure that this comparison is clearly only about the catalyst we studied here.

The authors have now included a Figure in SI (S16 - unless mistaken, it was not included in the first version of the manuscript) indicating that activity is independent of pressure above 5 bars for the catalyst bound to the amine ligand. This indeed is indicative of a complete displacement of the amine by ethylene. As in my first review, I find this point surprising, as it was clearly not observed by Jordan, Nozaki or Claverie in the past. For example, see entry 11-15 in Table 1 of *Organometallics* 2007, 26 (26), 6624–6635 or Table 2 in *Macromolecules* 2009, 42 (18), 6953–6963. Note that in *Organometallics*, 2012, 3128 (a paper cited in support by the authors), the catalyst is bound to DMSO or to a phosphine oxide. Not surprisingly, such weak ligands are easily displaced by ethylene. The authors should therefore explain why their amine bound to catalyst is so easily displaced by ethylene (could it be the difference in nucleophilicity between a tertiary amine and pyridine/lutidine)?

Response: Indeed the difference in nucleophilicity is the culprit for the higher lability of the tertiary amine versus. This is well established in the community, for example, the pyridine complexes mentioned by the reviewer are synthesized by adding a few equivalents of pyridine to displace the tertiary amine (TMEDA) of the starting material. An additional statement in our manuscript regarding the motivation of using NR3 as a ligand is now included.

Another point which should be clarified by the authors is whether this point relevant in BCP emulsion. At a same pressure, how much ethylene is dissolved in the BCP/water system compared to toluene? The authors did not properly address my point on mass transport limitation, most notably in an emulsion process. The argument brought by the authors is very weak (2g of PE in 15 minutes vs 2.6 g in 30 minutes). Considering the very complicated diffusional path of ethylene (from gas phase to water, then from water to various compartments of BCP and finally to the catalyst), the authors must consider whether physical processes, and not only chemical processes, do play a role in their kinetic observations. For example, essential reaction parameters such as catalyst concentration, number of BCP particles and stirring rate should be considered.

Response: We apologize for addressing mass transfer limitations only in the organic solution in our first response. Mass transfer limitations in the emulsion system were ruled out by demonstrating that varying the catalyst concentration in the micelle and changing the micelle concentration (independently) had no effect on the TOF. We modified our statement to prevent any misunderstanding regarding TOF definition.

The use of an equilibrium constant in equation 2 remains WRONG. The authors' justification (olefin complex not observed) is nonsensical. Spectroscopic studies are performed under conditions which are very different from kinetic experiments. I also note that such claim contradicts the surprising finding (see above) that the catalyst is saturated with ethylene. Surely, if the catalyst was so easily saturated with ethylene, the olefin complex would have been observed or trapped. Even worse, to back-up their (wrong) kinetic model, the authors claim that it was also used in *Macromolecules*, 2009, 6953. It is not the case. A conventional steady-state approximation is used (the term $k^{-1}L$ was present in the kinetic development, see SI). Last, the mathematical derivation of equation 3 in their refutation is also very

dubious. What is L? the amine? the added ligand? the sum of both? L is not a constant term (Total L = Bound L + Free L).

Response: [L] corresponds to the concentration of the additive. The polymerizations were performed with a large excess of additive versus catalyst concentration (e.g. for water at least ~1280 equiv) and therefore the effect of bound L is minimal in comparison to the free flowing [L].

Two additional points: How was the competition experiment performed with water? The concentration of water in toluene is miniscule. Table 4 should include units and experimental conditions (which catalyst, pressure and temperature).

Response: Table 4 reaction conditions are available in the supporting information.

Reviewer #2 (Remarks to the Author):

Reviewer #3 (Remarks to the Author):

With consideration of the author responses to all reviewer comments and the additional experimental data to test the possible role of water in chain transfer/termination, I support publication of this revised manuscript in Nature Comm.

Reviewer #4 (Remarks to the Author):

I was asked to comment on the authors' replies to reviewer 1 regarding the kinetic and mechanistic studies performed. My comments are limited to this section and I did not evaluate the manuscript in total. Reviewer 1 required clarification/correction of several points of the kinetic/mechanistic analysis. In my opinion the authors did not or only unsatisfyingly answer the major part of his questions. Alternatively they apologized for an unclear representation, but did not undertake any changes in the manuscript to clarify their presentation. On another occasion, they misquoted a publication in the rebuttal nearly to its opposite. Although I find the quality of their replies unsatisfactory to unacceptable, I have to side – for the most part – with the authors regarding their kinetic analysis. With one exception (equation (2) in the manuscript, see below), their analysis is correct and suitable for publication and I have to refute the reviewer's criticism.

In more detail:

1. Presumed zero-order in catalyst concentration

The reviewer was surprised by the zero order in catalyst concentration observed in table 3 which he correctly considers unlikely or indicating mass transport problems. The authors did not reply to this.

I suspect that the reviewer misunderstood the authors' representation. The turn-over-frequency (TOF) in table 3 is presented as "mol of ethylene consumed per mol of Pd per hour". Defined as such, the

expected linear dependence on Pd concentration should yield a constant TOF, which the authors indeed observed. The reviewer was probably misled by the authors' statement "This constant activity at various [...] catalyst loading suggests that the polymerization is not mass transfer limited." A slightly different expression, e. g. "The constant TOF per Pd at various...", might avoid this misunderstanding.

Response: Thank you for the suggestion. We have now rephrased our statement in the manuscript and implemented the reviewer's suggestion.

2. Zero-order concentration in ethylene concentration, possibly indicating mass transfer limitations
The reviewer questioned the possibility of saturation behaviour in Pd catalysed polymerization and raised the question whether the latter might not indicate mass transfer complications.

I concur completely with the authors' reply that there is no inherent reason to exclude saturation behaviour, i. e. an ethylene coordinated Pd complex becoming the catalyst resting state, for these complexes. The authors convincingly refer to the 2009 Mecking paper (ref. 65) where the same catalyst indeed showed saturation of activity above 5 bar ethylene. The fact the authors were able to obtain higher amounts (mass) of PE at the same ethylene pressures using different catalyst or catalyst loadings likewise indicate that mass transfer limitations are not a problem.

3. Inhibition effects

I concur with the authors' replies to the reviewer's comment.

However, this section required more than one reading to follow the authors' arguments. Unfortunately, although the authors apologized for the lacking clarity, they failed to make any changes to the text. A concluding remark that inhibition by PEG is the most likely culprit before the final conclusion, would be helpful. Likewise, the authors report that acidification enhances activity due to removal of amine, but they did not comment that the observed increase is trivial compared to the reduction of activity going from toluene to micelles. The authors did not discuss the influence on ethylene concentration in the micelles on activity, until the conclusion.

Response: A concluding remark that inhibition by PEG is the most likely culprit was added in the manuscript. The solubility of ethylene in the micelle is already mentioned in the main document. This discussion is highlighted by figure 4-b showing that higher ethylene pressure gives a higher rate. A comment was added to this existing section to help the reader. Finally, a statement in the manuscript was added to describe the trivial increase in activity under acidified condition.

4. Steady-state vs. pre-equilibrium approach

The reviewer remarks that the authors should apply a steady-state approach, comparable to Michaelis-Menten kinetics, rather than a fast pre-equilibrium approach. He states, correctly, that a pre-equilibrium approach is only justified if " k_p is small compared to k_{-1} ".

The authors state that an "olefin complex has never been observed for this catalyst". This forms the basis for a steady-state approach, but does ****not**** validate the further assumption of a fast pre-equilibrium, which relies on the relative rates of ligand attack on 2 vs. ethylene insertion. Simply stating that the same approach has been used previously likewise does not provide an answer to the reviewer's

criticism. In this context, one should not that the authors cite the 2009 Claverie Macromolecules paper as having previously used a pre-equilibrium approach. I looked up this article and found that Claverie started from the steady-state assumption the reviewer proposes, not from a pre-equilibrium assumption as the authors claim. Even more, Claverie et al state explicitly: "Contrarily to our initial belief, the insertion step is **not** preceded by a fast pre-equilibrium of ethylene coordination: both reactions go at comparable rates".

I believe, however, that the authors' approach of a pre-equilibrium is the better one in this case, for reasons they did not discuss: In the two extremes of the steady-state approach, we can assume that $k_{-1}L \gg k_p$ (i. e. a pre-equilibrium). In this case, the observed rate constant would be given by $k_{obs} = K[C_2H_4]k_p/[L]$. In the other extreme, $k_{-1}L \ll k_p$, we obtain $k_{obs} = k_1[C_2H_4]$, independent from $[L]$ (propagation rate determined by ethylene coordination, the intermediate trapped by immediate insertion). For $k_{-1}L \approx k_p$, we expect a more complicated behaviour with an apparent reaction order in L between 0 and 1. The fact that the authors observed a well-defined reaction order of -1 for the concentration of the inhibiting ligand L , indicates that k_p is indeed small compared to $k_{-1}L$. In this case a pre-equilibrium approach is to be preferred over a steady-state assumption, since it more correctly describes the behaviour at high monomer concentration.

Response: Our statement for the kinetic model we used was rephrased to highlight the hypothesis we made (fast preequilibrium) and a statement regarding the good agreement between this assumption and our experimental data was included.

5. Reviewer : "Furthermore, the passage from equation 2 to 3 is invalid. TOF and TONs are calculated using the TOTAL catalyst concentration. But in equation 3, the TOF was obtained by using $[1]$ in equation 2 instead of the total catalyst concentration."

The authors did not reply directly to this criticism but simply provided their derivation of equation (3). As the authors show in their answer, equation 3 is indeed correctly obtained considering that $[Pd]_0 = [1] + [2]$. Otherwise the term " $+ 1/k_p$ " would be missing in equation 3.

However, the reviewer is indeed correct that equation (2) **is wrong in this sequence**. In fact, no equation identical to or a derivative of equation (2) appears in the authors' rebuttal. As the authors show themselves, substitution of $[1]$ by $[Pd]_0 - [2]$ has to take place, **before** this is used to replace $[2]$ in $R_p = k_p[2]$. Equation (2) thus needs to be replaced by the $R_p/Pd_0 = \dots$ equation (or a derivative thereof) from their rebuttal. It would be very useful to include the full derivation in the supporting information.

Response: Thank you for pointing out that equation (2) did not match the full derivation we provided in our first response to the reviewer. We initially used equation (2) to reach equation (3) using a different substitution pathway as the one included in our first response to the reviewer. The full derivation is now included in the supporting information. We also included an additional equation defining the term TOF.